# An Analysis of Theoretical Perspectives in Research on Nature-Based Interventions and Pain

**DOI:** 10.3390/ijerph191912740

**Published:** 2022-10-05

**Authors:** Reo J. F. Jones, Chloé O. R. Littzen

**Affiliations:** 1School of Nursing, Oregon Health & Science University, Portland, OR 97239, USA; 2School of Nursing and Health Innovations, The University of Portland, Portland, OR 97203, USA

**Keywords:** nature-based interventions, pain, nursing theory, theory, stress reduction theory, attention restoration theory, biophilia hypothesis, greenspace

## Abstract

Chronic pain results from a complex series of biomechanical, inflammatory, neurological, psychological, social, and environmental mechanisms. Pain and pain-related diseases are the leading causes of disability and disease burden globally. Employing nature-based interventions for the treatment of pain is an emerging field. Current theory driving the suggested mechanism(s) linking the pain reducing effects of nature-based interventions is lacking. A two-step approach was taken to complete a theoretical review and analysis. First, a literature review was completed to gather a substantive amount of research related to theoretical frameworks on the topic of nature-based interventions and pain. Secondly, a theoretical analysis as proposed by Walker and Avant was completed to explore current theoretical frameworks accepted in the literature on nature-based interventions and pain. Stress reduction theory and attention restoration theory were the most common theoretical frameworks identified. Neither theoretical framework explicitly identifies, describes, or intends to adequately measure the concept of pain, revealing a limitation for their application in research with nature-based interventions and pain. Theoretical development is needed, as it pertains to nature-based interventions and pain. Without this development, research on nature-based interventions and pain will continue to use proxy concepts for measurement and may result in misrepresented findings.

## 1. Introduction

The International Society for the Study of Pain defines “pain” as an unpleasant sensory and emotional experience associated with tissue damage [1,2,3]. The experience of pain involves various complex pathways which have physiological and psychological implications [1,4]. Pain processing involves neural networks linked to the autonomic nervous system (ANS), through the sympathetic nervous system (SNS) wherein bundled neurons in the SNS (or ganglia) running along the spinal column receive pain signals [5,6]. Generally, there are two primary types of pain described in current medical literature: acute and chronic [7]. Acute pain may be induced by tissue injury and subsequent inflammation, a skeletal muscle spasm, or other complex nociceptive reactions to sympathetic nervous system activation [8]. Nociceptors, or nerve cell endings, initiate the perception of pain and relay the processing of “painful” stimuli to the central and peripheral nervous systems. In acute pain, the signals subside, and the relay turns off. However, the threshold for nociceptors to fire is reduced during inflammation or ongoing tissue injury, which can lead to heightened pain sensitivity (hyperalgesia) and chronic pain [8].

The transition from acute to chronic pain is associated with several physiological, psychological, and psychosocial predictors involving central nervous system (CNS) and peripheral nervous system (PNS) pathways involved in stress reactivity and activation of the hypothalamic-pituitary-adrenal (HPA) axis [9]. Chronic pain is often described as a state of stress, and the high incidence of stress, depression, and anxiety are hallmarks of pain-associated chronic illness [10,11]. The HPA axis plays a role in mediating the relationship between stress and pain [12]. It has been suggested that ongoing and unmitigated activation of the HPA axis and subsequent release of glucocorticoids is also a factor in pain chronicity and comorbidities of pain such as anxiety and depression [9]. Pain also adds stress to the human system, whether physical (homeostatic dysregulation) or emotional (perceived stress) [13].

### 1.1. Acute and Chronic Pain

Within this review, two types of pain are discussed—acute and chronic. Acute pain is defined as nociceptive pain caused by a specific injury or chronic illness (associated with SNS activation) and subsides after the stimulus ceases [7]. Chronic pain—the focus of this manuscript—is dynamic and results from a complex series of biomechanical, inflammatory, neurological, psychological, social, and environmental mechanisms [14]. Unmitigated, chronic pain can lead to treatment-resistant pain [15]. It is estimated that over 20% of adults within the United States (U.S.) suffer from chronic pain, and more than 8.0% adults in the U.S. have “high-impact” chronic pain, or persistent pain that can adversely impact daily life for six months or more [14]. Higher prevalence of chronic and high-impact pain is reported among women, older adults, and socially marginalized adults, as well as adults living in rural regions or on public assistance [14].

The Global Burden of Disease Study 2016 reported that pain and pain-related diseases are the leading causes of disability and disease burden globally [16]. Low back pain and headache disorders are among the top 10 causes of disability-adjusted life-years (DALYs)—a measure combining years of life lost due to premature mortality (YLLs) and years of life lost due to time living with illness or healthy life lost due to a disability (YLDs) [17]—amongst all age-groups [18].

Acute and chronic pain are different clinical states, but one can experience acute episodes of pain within a chronic pain disorder [15]. Chronic pain is not self-limiting and can arise from the additional burden of psychological distress associated with ongoing episodic issues of acute pain outlasting the healing process. It is also considered a disease state associated with pain-inducing chronic illness [15,19]. Left unmitigated, chronic pain can lead to complex pain disorders, such as chronic widespread pain or fibromyalgia, and associated conditions such as depression, anxiety, and psychological stress [20].

### 1.2. Nature-Based Interventions

For the purposes of this theoretical review and analysis, and to incorporate a working definition of nature-based interventions (NBIs) accepted in current literature, we propose the following: A comprehensive working definition of “nature” by researchers of nature contact and human health Bratman et al. (2021) will be used within this paper, whereas “nature” means properties that include various aspects of outdoor landscapes; and, that these landscapes are “encompassing elements and phenomena of Earth’s lands, waters, and biodiversity, across spatial scales and degrees of human influence, from a potted plant or a small urban creek or park to expansive, ‘pristine’ wilderness with its dynamics of fire, weather, geology, and other forces” [21]. NBIs are “nature-based”, such that they involve actual or simulated elements of nature, incorporating interventional strategies involving greenspaces—i.e., grassy fields, forested settings, parks, vistas, green foliage, and plant imagery, blue-spaces such as waterfalls or streams, and brown-spaces such as savannahs and desserts—all within the interventional setting [22,23].

One emerging field of study employing NBIs for health and wellness is on the connection between nature exposure and the mitigation, or treatment, of pain, stress, and the burden of chronic illness [24,25,26,27]. Several studies describe the pain-reducing effects of viewing “live” nature, as well as simulated nature, in the form of pictorial or virtual images of natural landscapes such as greenspaces in clinical settings for improved health and well-being, including reducing pain outcomes [27,28,29,30,31,32]. For example, early research demonstrates that exposure to greenspaces may reduce pain and its chronicity for individuals with arthritic conditions [27,33,34,35,36].

Potential pathways of pain reduction include evidence suggesting that forest aerosols, from common aromatic plants found in forests and greenspaces, may reduce inflammation and pain [37,38,39], especially when tested in environmentally controlled settings [40,41]. It is also known that simulated audio-visual nature scenes can reduce stress reactivity, improve PNS response, and decrease pain perception [32,42]. However, the theory driving the suggested mechanism(s) linking the pain-reducing effects of different types of NBIs, such as Shinrin Yoku, Greenspace Interventions, or simulated nature contact, and for whom those potential effects are most beneficial, is lacking.

In the discipline of nursing, theories are traditionally considered “organizations of concepts and evidence into conceptual structures that help practitioners and researchers see pattern and organization in their activities and make sense of what they observe and discover in the world” [43] (p. 7). Alternatively, theory is considered an “explanation of what is going on” [44] (p. 2). As it pertains to scholarly work, theory is the scaffolding upon which we build our knowledge. Without theory, we are unable to determine where we have been or where we are going. Theory enables us to build upon the important work of those that came before us and help us determine what roads have been left uncharted. To date, there has been a paucity of theory development focused on nature-based interventions in nursing. Outside of nursing, theories focused on nature-based intervention have been developed, but there has been general acceptance by application of these theories without critique of their adequacy for research focused on pain. Therefore, the purpose of this article is to complete a theoretical review and analysis, otherwise referred to as a theory critique, of the currently accepted theories within research on nature-based interventions and pain.

## 2. Materials and Methods

A two-step approach was taken to complete this theoretical review and analysis. First, a literature review was completed to gather substantive literature related to theoretical frameworks on the topic of nature-based interventions and pain. Secondly, a theoretical analysis was completed to explore the relevance of current theoretical frameworks accepted in the literature for application to future research on nature-based interventions and pain in the discipline of nursing.

### 2.1. Literature Review

A literature search was conducted from August 2020 to April 2022 via the databases PubMed, CINAHL, Scopus, and ScienceDirect. Key terms included “pain” and “greenspace” or “forest bathing” or “forest therapy” or “shinrin-yoku” or “nature therapy” or “nature-based intervention.” Inclusion criteria consisted of articles (a) in the English language; (b) peer-reviewed; (c) either guided by an applied theoretical framework or discussed theoretical concepts and hypotheses; (d) was an intervention study; and (e) in the discipline of nursing or psychology. The rationale for including articles from the discipline of psychology was because this is where the primacy of research on nature-based interventions has been completed. For the purpose of this integrative review, all levels of theory except for conceptual models and hypotheses were included in order to focus on the more conventionally accepted theories related to nature-based interviews and pain. Conceptual models, or hypotheses related to our findings will be discussed to advance theoretical thinking in the discipline of nursing. No time restrictions on articles were selected to provide a comprehensive overview.

### 2.2. Theory Analysis

The six-step method of theory analysis by Walker and Avant (i.e., origins, meaning, logical adequacy, usefulness, generalizability and parsimony, and testability) was applied to critique theoretical frameworks on the topic of nature-based interventions and pain [45]. According to Walker and Avant, analysis enables us to determine if a theory is valid and reliable while revealing its strengths and weaknesses. Examining the current state of theories within nature-based interventions and pain will also identify gaps for further theoretical development in nursing.

## 3. Results

Twenty-eight articles were in the initial literature review. A total of 19 articles were removed as they did not meet inclusion criteria for the literature review in this theoretical analysis. These included studies that were either non-experimental observational, studies which did not describe or apply a theoretical foundation, or studies published earlier than 2010, resulting in a total of nine articles (see Table 1). From the remaining nine articles, a total of three focused on nature-based interventions and pain and applied a theoretical framework. One of these articles applies a theory unrelated to the concept of nature and pain. Across the nine articles, two substantive theories were demonstrated: Stress Reduction Theory (SRT) and Attention Restoration Theory (ART) [46,47]. Two of the articles mentioned Wilson’s Biophilia Hypothesis [48].

No other theoretical frameworks were identified for the purpose of this review and critique. Some of the 19 articles which were originally removed are described in the background and discussion sections of this paper as they include reviews on nature contact and health outcomes pertaining to pain research, in addition to literature reviews on the theoretical foundations of nature contact and health outcomes.

The following paragraphs will focus on the theory critique of both SRT and ART, as it was demonstrated they were the substantive theoretical frameworks for nature-based interventions and pain, as well as a review of Biophilia as an interrelated hypothesis. To clearly indicate which articles either described a theoretical framework or applied theory in the development of their study and as a foundation for their analysis, we use the language “described” or “applied” in the column “Theoretical Framework.” The details of these articles, and their application of theory, are further summarized in this section.

### 3.1. Stress Reduction Theory

Roger Ulrich, an architect, and health science researcher credited for incorporating elements of the natural world such as foliage and views of greenspaces into evidence-based healthcare design, deductively developed the Stress Reduction Theory (SRT) during the 1980s and early 1990s [56]. Ulrich was inspired by the city planner and landscape architect Frederick Olmstead who wrote about the impact of viewing nature and its capacity to engage the individual without fatigue, to relieve the stresses of city life, to calm and yet energize the mind, and thusly, the body, and therefore nature was viewed as not only an antidote to stress but necessary to preserve and mold for leisure [57,58] The “nature” which Olmstead described was that of the mid-19th century Americas where urbanicity was just beginning to expand westward and much of the nature surrounding burgeoning cities was relatively pristine and organic.

Ulrich had a keen understanding of “overload” and “arousal,” theories stemming from the social and natural sciences of the 1960s and 1970s, which he broadly described as theories encompassing the concept that visual complexity, noise, intensity of movement, high levels of stimulation (e.g., being immersed in a bustling city), can overwhelm, and fatigue the human “perceptual system,” and lead to high levels of stress. Ulrich further noted that these theories implied that restoring one’s pre-stressed state was possible through passive engagement with the natural world and learned positive associations with elements of nature (e.g., plants, parks, wilderness, etc.). Early research into these psychological theories comparing physiologic excitement between urban and natural settings [56,58] is an interventional strategy from which much research on nature and health stems [59].

Ulrich pioneered studying the impact of the environment in healthcare settings on stress recovery, which inspired the development of his theory. In 1984, he studied post-surgical patients viewing scenes of nature through their hospital windows, and found that they recovered quicker, and were discharged sooner, than patients without such views of nature [31]. Ulrich posited that the observed stress recovery may be due to a psycho-evolutionarily based aesthetic and affective predisposition for natural scenery [58,60] The seminal work from which SRT originated involved a study wherein Ulrich and colleagues exposed 120 healthy undergraduate adult subjects (60 male and 60 female) in the United States to a stressful filmed event, then immediately exposed the stressed subjects to videos of nature or urban settings. Participants’ physiologic and psychologic stress recovery was faster for participants viewing nature videos than for those viewing urban settings [56]. Therefore, SRT stemmed from Ulrich’s extensive research on the hypothesis that outdoor or “natural” environments are less threatening and stressful than urban or built environments.

Within SRT, there are three primary concepts of focus—although they are not explicitly stated or visually modeled—stress, a natural environment, and an urban environment. Ulrich often described “stress” as it has been defined by environmental psychologists and researchers, who described the connection between environmental stress and health outcomes through pathways associated with evolutionary theory [56,60,61]. Therefore, in studies pertaining to stress recovery or reduction, Ulrich denotes stress as a process involving an individual’s psychological (cognitive appraisal of the stressor and subsequent emotional response), physiological (engagement of the nervous system and subsequent cardiovascular, neuroendocrine, and musculoskeletal systems), and behavioral (what an individual does to cope, or navigate the stress reaction) response to any situation or environment which threatens or challenges the individual’s well-being [56].

While Ulrich did not explicitly define nature in the development of SRT, within Ulrich’s work, “nature” is described as an outdoor environment or landscape with some inherent complexity consisting of “vegetation,” such as trees and flowers, and/or “water,” such as rivers and lakes, whereas “urban” environments consist of the “built” world, or cityscapes with limited nature [46]. It should be noted that Ulrich described nature in contrast to the built city environment, but natural environments could also be shaped by humankind (e.g., “pastoral parks,” or total “wilderness”) [56]. For example, within the seminal study that informed SRT, Ulrich employed “natural” landscapes vs. “urban” landscapes such that “nature vegetation” consisted of a setting dominated by trees… other vegetation… occasional light breeze in background… or “water” consisting of “a fast-moving stream; waves and ripples…” in contrast to “urban heavy traffic,” which consisted of a “commercial” scene involving “fast moving cars” throughout a city [56].

Within SRT, it is postulated that in the presence of a natural environment, there is an inverse association with stress, meaning that stress decreases and promotes stress recovery. Comparably, the reverse is true in the presence of an urban environment; where there is a positive association with stress, meaning that stress increases (or remains unchanged) in an urban environment and delays or prevents stress recovery.

Independent of their meaning, the concepts and statements demonstrate logical adequacy within SRT. No logical fallacies within the structure of SRT were detected. Moreover, how the theory has been described has promoted ease in predictability and testing across decades of research and disciplines. Practically speaking, SRT has usefulness in the sense that it promotes ease in connecting concrete concepts (natural and urban environments) with a measurable abstract concept (stress). Across disciplines, it has been demonstrated how SRT is practical as it pertains to stress recovery of participants. In nursing, there is a paucity of demonstrable practicality in the literature, but theoretically speaking, SRT has the potential of practical and predictable outcomes within nursing science. The generalizability of SRT is sound, meaning that it is easy to interpret what will occur from implementing and testing the theory (e.g., with nature exposure one should expect a decrease in stress). For parsimony, SRT can be simply stated without the essence being diluted. SRT has been empirically supported across the literature for decades [24,56,59,62,63,64,65]. Moreover, SRT promotes the development of hypotheses that can be subjected to ongoing empirical research.

### 3.2. Attention Restoration Theory

The Attention Restoration Theory (ART) was developed by psychology professors and researchers Rachel and Stephen Kaplan in the 1980s [47]. ART proposed that attention can be described in two ways, (a) voluntary-directed, focused, cognitively controlled attention, or (b) involuntary, where attention can be unconsciously directed to meaningful stimuli, such as elements of the nature, or the natural world [66,67,68]. According to Stevenson et al. [69], Kaplan and Kaplan applied the term “directed attention” to distinguish themselves from previous work on the concept of voluntary attention [70]. ART also suggests that involuntary attention can be improved by spending time in nature, “restoring” the cognitive capacity for focus, and ultimately reducing psychosocial stress. ART is rooted in four key properties described by Kaplan and Kaplan. One property is the role of “soft fascination,” how humans view nature as meaningful, such that aspects of the natural environment can capture attention effortlessly which has been correlated with perceived restorativeness in present day research on the calming effects of nature [47,71]. Other properties of ART include the extent to which one is immersed in the natural world, the concept of “being away,” from usual daily activities, and an individual’s compatibility with their environment (e.g., being exposed to a natural setting that one appreciates) [47,68]. Kaplan and Kaplan posited that the cognitive capacity for focus can be depleted but redirecting attention to nature can restore this cognitive capacity and increase attentional focus [47,68].

The primary concepts within ART include nature exposure (also referred to as a restorative environment), and attention (involuntary and/or directed). Nature exposure is defined as being exposed to a natural setting that one appreciates [47,68]. Involuntary attention is defined as where attention can be unconsciously directed to meaningful stimuli, such as elements of nature, or the natural world [67,68]. It has been reported in current research there is ambiguity around the concept of directed attention [69]. While not explicitly stated, it is implied that nature exposure is positively associated with attention, meaning that attention will increase with the presence of nature exposure.

Independent of their meaning, the concepts and statements demonstrate logical adequacy within ART. How the theory has been described has promoted some ease in predictability and testing across decades of research and disciplines, but it can be argued due to the complexity of the theory, as it is explained, some concepts appear to be supplemental and not intentionally measurable (e.g., directed attention) within empirical research. Moreover, both nature exposure and attention as concepts have been reported to be used interchangeably with other concepts (e.g., attention with concepts such as mental fatigue, concentration, or executive function) adding further confusion. Practically speaking, ART has usefulness in the sense that it promotes ease in connecting relatively concrete concepts (nature exposure) with a measurable abstract concept (attention). The generalizability or ART is somewhat sound, meaning that it is easy to interpret what will occur from implementing and testing the theory (e.g., with nature exposure one should expect an involuntary attention and thus a reduction in stress). For parsimony, ART can be stated simply; the addition of directed attention promotes unnecessary complexity and more so is applied in variable ways across empirical research [69]. ART has been empirically supported across the literature for decades [69,72]. Moreover, the SRT promotes the development of hypothesis that can be subjected to ongoing empirical research, although conceptual variability promotes limitations in testing.

## 4. Discussion

In our theoretical review, the most applied theoretical frameworks within nature-based interventions and pain research were SRT and ART. Through the theoretical analysis, we demonstrated that both SRT and ART are sound theoretical frameworks based upon the criteria proposed by Walker and Avant [45], yet a gap was identified relating to the concept of pain. Specifically, neither SRT or ART explicitly identify, describe, or intend to measure the concept of pain. This is of concern from a standpoint of scientific rigor as scientists applying these theories are using the concepts of attention or stress as proxy variables for pain. Additionally, scientists applying concepts of attention or stress to pain need to ensure that the psychologic and physiologic association between these different physical states and processes are well-defined, or that any overlap specific to the psychologic and physiologic pathways connecting attention, stress, and pain are explicitly defined and reflected within the research design, measures, and outcomes. This can be interpreted as a concern for construct validity, meaning that scientists are not actually measuring what they intend to measure [73]. From the patient perspective, developing and testing interventions for pain reduction (whether acute or chronic) without supportive knowledge or accurate dissemination is inherently unethical, though the intention to support pain reduction is caring-focused and a critical focus of nursing science and health care. Future theoretical development is needed to demonstrate the specific connection between nature-based interventions and pain, including the different pain pathways that may be impacted, as well as how these pathways are associated (i.e., psychological and physiological). Further, key suggested mechanisms inherent to nature-based interventions and which aspects of these interventions offer support for pain mitigation should be a primary focus analyzed within the literature [24,27].

Exposure to natural landscapes to improve health outcomes on the basis of SRT and ART has been the subject of much research, especially in regard to measuring cognition; emotionality; psychological well-being; managing symptoms of chronic illness [64,74,75,76,77] mitigating acute [30,52] and chronic pain, respectively [25,51]; attentional capacity as improved by exposure to nature [63,66,78] and the human capacity for perceiving restorativeness as a construct [79,80]. SRT and ART are paramount as they provide the underlying framework supporting the notion that contact with nature can decrease psychological and physiological stress, restore cognitive focus, and increase feelings of relaxation. SRT and ART are comprehensive theories; they can provide a conceptual basis for studying a variety of interventional strategies and health outcomes associated with the psychology and physiology of stress and well-being, although ART has some limitations as aforenoted.

Various types of exposure to nature (such as residential greenspace) may produce an “affective” benefit, such that moods, feelings, emotions, and/or stress responses can be positively impacted by greater contact with nature [81,82,83,84]. A myriad of potential mechanisms and effect modifiers may impact these benefits which is beyond the scope of this paper [82]. Exposure to nature has been shown to improve pain perception, but the specific mechanisms underlying this observed benefit remain somewhat unclear and are of primary focus in current research [27,28,85].

Of the articles included in this review, a total of three out of nine applied theory within their study backgrounds and design and offered some discussion of how these theoretical frameworks support the association between nature and pain reduction [29,52,53]. For example, Tanja-Dijkstra et al. [52] discussed a theory not analyzed in this article called the Elaborated Intrusion Theory [52], which they correlated with SRT in the sense that viewing nature inspires positive associations—and these positive associations can “intrude” a person’s thoughts, increasing the “value” of the nature-image and distracting from pain. The notion of “distraction” is somewhat related to the applied theories within Wells et al. [53] and Scates et al. [29], wherein ART is incorporated to explain how inhibitory mechanisms involved in directed attention could be “restored” by exposure to nature through soft fascination [53], and this improvement of directed attention invites a “positive distraction” from the experience and sensation of pain [29]. Scates et al. [29] additionally noted that SRT supported the notion that nature scenery reduces stress, which is a central pathway toward improving directed attentional focus. These conceptual and linguistic overlaps suggest that not only are there some emergent pathways linking these foundational theories, but that they need to be distinctly understood and clarified.

Each of the remaining six articles described ART, SRT, or aspects of the biophilia hypothesis within their study’s backgrounds [28,49,50,51,54,55]. For example, Verzwyvelt et al. [54] cited a study by Ulrich [86], which called for more nature and daylight exposure in hospital environmental design but did not explicitly discuss this literature in reference to the “biophilic environments” within their study [86]. In contrast, Verzwyvelt et al. [54] focused on Wilson’s biophilia hypothesis described in the next paragraph [87]. In Li et al. [28], authors provided some background on previous theories in pain research and gave considerable credence to the notion that there are major gaps linking pain relief to greenspace exposures, specifically the ideas of symptom distraction through redirected attention, immune-modulating phytoncides from exposure to trees, and negative ions in the air, among other concepts linking greenspaces to health. Li et al. (2021) also suggested that the pathways linking research on pain and nature to improved health outcomes is thus far relatively indirect, and while they do not cite a specific theory, they do suggest further study to bolster early research on the subject, as within Ulrich [31]. Lipponen et al. [55] also suggest that ART and SRT provide background pertaining to mechanisms and pathways associated with the benefits of nature-based interventions, but do not specifically integrate these theories into their study. Han et al. [51] cite ART within their reference list and suggest that mechanisms of forest therapy (related to Shinrin Yoku or forest bathing) may induce physical relaxation and feelings of “restoration” alongside activation of the parasympathetic autonomic nervous system for improved pain outcomes in chronic pain sufferers. Ali Khan et al. [50] also cited the work of Ulrich and the benefits of viewing nature for pain relief [31], but do not go into detail on the theory. Rather, their study emphasizes facets of ART and SRT, including how direct or indirect interaction with nature (in the form of plants and flowers) can improve mental health by decreasing anxiety, stress, and depression—comorbidities of pain; and such interaction with nature may also distract from the pain experience. Lechtzin et al. [49] also discuss the work of Ulrich and SRT and Ulrich’s seminal study on nature and pain in hospital settings [31], which inspired their nature-image selection for their study of nature’s calmative effects. While these six studies reference the work of ART, SRT, and biophilia with variable detail, they also highlight the fact that there are significant gaps connecting NBIs, theory, and pain research for health and well-being.

### A Note on Biophilia Hypothesis

Although we did not include biophilia hypothesis in our theoretical analysis, we find it important for the reader to provide some background on this perspective. In 1986, biologist, naturalist, and author, E.O. Wilson defined “biophilia” as the “innate tendency to focus on life and life-like processes” [48] (p. 1). The biophilia hypothesis therefore suggests that not only do humans have an innate affinity for all living things such as the natural world, but also that this affinity is rooted in our personal evolutionary history, [88]. As human evolution occurred through interaction with the physical environment, or the natural world, it has been suggested that humans have a biologically based biophilic tendency to achieve a state of well-being in nature [89,90], as opposed to the urban and indoor environs, which are described as more stress-inducing in the current literature on biophilic design [48].

Criticism of biophilia as a working hypothesis is largely of semantic origin. Critics argue that the definition of biophilia is too broad, that there is a lack of research on the evolution of these biophilic genetic predispositions, or that the natural environments of which biophilia is concerned are too vast and varied [88]. However, biophilia as a hypothesis is open to change and development—E.O. Wilson wrote about it and its complexity over several years [48,91]. Biophilia, when applied to interventional research design, has been associated with significant positive health benefits [48,87,91,92]. In our literature review investigating experimental studies involving NBIs and their impact on pain, 2 out of 15 studies referenced biophilia as a direct influence on their study designs [49,54].

In Verzwyvelt et al. [54], a “biophilic” VR environment was used alongside a standard treatment “control” environment to decrease pain and stress during chemotherapy infusions for 33 adults with active cancers. Authors described their logic for designing the biophilic virtual environment in their study based on the idea that such an environment elicits innate positive associations between humans and elements of nature such as greenery, garden plants, and vistas of natural landscapes to improve mental health and overall patient outcomes [54]. Lechtzin et al. [49] randomly exposed 120 adults to one of three environmental conditions in an effort to determine the impact of nature’s sights and sounds on pain during a bone marrow aspirate procedure. Participants in the “nature arm” of the study were exposed to conditions that the researchers described as specifically designed based on biophilia and the work of Wilson—a “pastoral” scene of a natural landscape with foliage, water, and skyline along with paired nature sounds [49]. While neither study yielded significant results in pain reduction initially, the study by Verzwyvelt et al. [54] demonstrated patient satisfaction with the biophilic environments such that they were “enjoyable” and “fun.” Participants in the study by Lechtzin et al. [49] reported higher overall satisfaction with their procedure in the “nature arm” group. However, it is unclear to what extent biophilia influenced the results in these two studies, further bolstering the need to incorporate greater specificity in research linking concepts of biophilia directly with pathways involved in improving health outcomes.

As mentioned, the field investigating the impact of NBIs on pain and comorbidities of pain is growing. To fill theoretical gaps, it is essential that further research links the proposed mechanism(s) driving pain reducing effects of NBIs, and for whom those potential effects are most beneficial. Based on the findings of our review, some studies applying NBIs for pain relief either do not reference theory, or apply theories centered on the experience of pain without mentioning the inclusion of “nature” in the interventional design of the study [52]. In a recent integrative review, Stanhope et al. [27] proposed a conceptual framework linking greenspace to improved pain outcomes. Authors described greenspace exposures as encompassing increased potential for activity, increased sunlight, sociality, negative ions in the atmosphere, exposure to nature sights, nature, sounds, phytoncides or forest aerosols, and even the microbiome, alongside several proposed mechanisms of action linking these exposures to pain relief. Mechanisms of action, referred to as “ecophysiological linkage mechanisms,” included physiological and psychological facets of pain biology (e.g., stress, sleep, mental health, nociception, immune mediators, etc.) as pathways between greenspace exposure and improved pain outcomes [27] (p. 4). Based on the findings of this review, the work of Stanhope et al. [27], and emergent research on the field of nature-based interventions for improved pain outcomes, we believe a novel conceptual framework should be developed, applied, tested, and retested to formulate a working grand theory to support future research.

## 5. Strengths and Limitations

The narrow inclusion criteria for article selection in this theoretical review and analysis can be seen as both a strength and a limitation. As a strength, the narrow inclusion criteria promote a foundational understanding of the theoretical frameworks most often adopted or applied in nature-based intervention research and pain. Comparably, this narrow inclusion criteria can also be a limiting factor potentially peripheralizing theoretical frameworks that have been less widely adopted and applied. It is our hope that with this foundational theoretical review and analysis, scholars can use this article to continue to build an understanding of the theoretical frameworks potentially less widely accepted. Additionally, there is a dearth of theoretically grounded experimental literature on nature contact as an interventional strategy for pain mitigation or management. In this analysis, we aimed to highlight the theoretical underpinnings and working hypothesis informing this literature to date. A result of this endeavor was the discovery of a relatively limited number of articles investigating the impact of nature-based interventions on pain outcomes.

## 6. Conclusions and Future Theory Development

The literature review revealed that stress reduction theory and attention restoration theory are the most discussed and applied theoretical frameworks when studying nature-based interventions and pain. In analyzing both these theories, it was determined that neither identify, describe, or adequately measure the concept of pain, revealing a limitation for their application in research with nature-based interventions and pain. Theoretical development is needed, both within the discipline of nursing and externally, as it pertains to nature-based interventions and pain. Without development of theory focused on pain, knowledge production will focus on using proxy concepts to measure pain which may lead to inconsistent or inaccurate findings in nature-based intervention research on pain.

## Figures and Tables

**Table 1 ijerph-19-12740-t001:** Articles included in the theoretical analysis based on the literature review.

Article Name, Authors, Year, and Country	Sample	Aims	Method/Design	Theoretical Framework
A Randomized Trial of Nature Scenery and Sounds Versus Urban Scenery and Sounds to Reduce Pain in Adults Undergoing Bone Marrow Aspirate and Biopsy, Lechtzin et al., 2010, U.S.A. [49]	N = 120 adult patients Nature arm (n = 44), City arm (n = 39), standard care arm (n = 37).	Determine the impact of exposure to sights and sounds of nature on pain outcomes during Bone Marrow Aspirate and Biopsy (BMAB).	RCT with 3 groups. Groups were exposed to an audio-visual nature scene, city scene, or were provided standard care during the procedure. Pain scores and categorical pain outcomes were measured.	Described
Plant Therapy: a Nonpharmacological and Noninvasive Treatment Approach Medically Beneficial to the Wellbeing of Hospital Patients, Khan et al., 2016, Germany [50]	N = 270 adult patients	To investigate the effect of therapeutic horticulture on health outcomes in patients within two surgical wards of a hospital.	Mixed methods. Patients were randomly assigned to either ward with a total of 135 patients in each group. In ward A, patients were exposed to therapeutic horticulture, and in ward B, patients were exposed to standard hospital rooms. Small group discussion and focal interviews followed the intervention. Vital signs, hospital stay (days), and analgesic consumption were measured.	Described
The Effects of Forest Therapy on Coping with Chronic Widespread Pain: Physiological and Psychological Differences between Participants in a Forest Therapy Program and a Control Group, Han et al., 2016, South Korea [51]	N = 61 adults Control (n = 28) or FT (n = 33)	Test the impact of forest therapy (FT) sessions on symptoms of chronic widespread pain (CWP).	Quasi-experimental, two-groups repeated measures design. FT sessions lasted 2 days and were designed with physical activities and psychological approaches to address CWP. Measures included Heart-rate Variability, with NK cell activity, pain and depressive symptoms, and health-related quality of life.	Described
The Soothing Sea: A Virtual Coastal Walk Can Reduce Experienced and Recollected Pain, Tanja-Dijkstra et al., 2017, U.K. [52]	N = 85 adults (Study 1) N = 70 adults (Study 2)	To investigate the impact of simulated nature via Virtual Reality (VR) on pain immediately after the VR (experienced pain), and recollected pain after one week posttest.	Quasi-experimental randomized between- participants design. Study 1 used a cold pressor test to induce pain, Study 2 was a randomized trial with patients undergoing a dental treatment. In Study 1 and 2, the 3D nature image used as the VR setting included a coastal environment with green landscape and foliage. Measures included experienced and recollected pain measured using the 0–11 numeric rating scale and the McGill Pain Questionnaire.	Applied
Nearby Nature Buffers the Pain Catastrophizing–Pain Intensity Relation Among Urban Residents With Chronic Pain, Wells et al., 2019, U.S.A. [53]	N = 80 middle-aged adults	To investigate the moderating effects of nearby nature on the association between pain catastrophizing and daily pain intensity, and the association between rumination and daily pain intensity	Quasi-experimental study of secondary data. Measures included proximity to nature, pain scores, measures of pain catastrophizing, rumination, helplessness, and magnification, alongside time spent in nature.	Applied
Using Nature-Inspired Virtual Reality as a Distraction to Reduce Stress and Pain Among Cancer Patients, Scates et al., 2020, U.S.A. [29]	N = 50 adults	To determine the impact of a virtual nature simulation on stress and pain in 50 adult cancer patients during their IV infusions and/or port access procedures (acute pain).	Mixed methods, repeated measures design. Measures included questionnaires developed by researchers focusing on constructs of stress and pain with open-ended interview questions.	Applied
Effects of Virtual Reality v. Biophilic Environments on Pain and Distress in Oncology Patients: a Case-Crossover Pilot Study, Verzwyvelt et al., 2021, U.S.A. [54]	N = 33 adults	To investigate the impact of using either a “biophilic Green Therapy or Virtual Reality” exposure environment as compared to a control, to decreasing pain and distress while participants received chemotherapy.	A crossover design pilot study with adult participants experiencing breast, gynecologic, gastrointestinal, pancreatic, and prostate cancers. Participants exposed to 3 settings over 3 different cycles, including a control room, a Green Therapy room, and a VR room to receive chemotherapy. Measures included pain, distress, heart rate, blood pressure, and salivary cortisol.	Described
Can Residential Greenspace Exposure Improve Pain Experience? A Comparison between Physical Visit and Image Viewing, Li et al., 2021, China [28]	N = 24 young adults	To evaluate the effects of two environments, outdoor greenspace, versus viewing a simulated greenspace, or a control environment (an empty room) on pain perception, pain threshold, and pain tolerance.	Quasi-experimental randomized cross-over design. Pain was induced via electrical pain stimuli. Measures included pain intensity, anxiety, two adjective pairs were used to measure the state of anxiety and subjective stress, as well as heart rate, heart rate variability and blood pressure. A measure of Scenic Beauty Estimation (SBE) was used to assess participants’ preference regarding the experimental environments.	Described
Effects of Nature-Based Intervention in Occupational Health Care on Stress—A Finnish Pilot Study Comparing Stress Evaluation Methods, Lipponen et al., 2022,Finland [55]	N = 11 middle-aged females	To assess methodologies on Nature-Based Interventions (NBIs) and their limitations for measuring psychological and physiological effects over time.	Quasi-experimental longitudinal pilot. The NBI included six group appointments over six months. Measures included heart-rate variability, self-reported pain, and work exhaustion measured pre and post study period.	Described

## Data Availability

Not applicable to the content of this review article.

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
