# Peer review of "An Analysis of Theoretical Perspectives in Research on Nature-Based Interventions and Pain"

_ijerph, 2022, doi:10.3390/ijerph191912740_

Round 1

Reviewer 1 Report

Dear Authors,

Thank you for submitting this interesting read on the topic of ‘nature-based interventions and pain’. In general, I find that the manuscript gives a nice introduction and basic overview to the topic of pain and the relevance of an analysis on research done on nature-based interventions. Also, the overall methodology of the paper, addressing the theoretical perspectives and research gaps, are well-chosen and for most parts well-structured and well-written.

Furthermore, your choice of topic is highly relevant, not just to nursing but in a broader perspective addressing for instance architectural design related healthcare environments. I would thus highly consider providing your final article to my students in one of my courses, as it provides a nice overview of the three perspectives of SRT, ART and Biophilia.   

However, I have a few, minor suggestions for you to consider helping improve the manuscript:

1)      Even though the authors quite clearly explain the structure of the paper, I am in doubt about the placement of the part on biophilia. At presents, it appears towards the end of the paper in section 4.1 as an important background. But it comes after the main content – the review and analysis. I think it would stand stronger, methodologically, if moved forward and deliberately used in the comparison/analysis of the perspectives of SRT and ART.  

2)      Finally, I am rather surprised with the limited number of articles in the review. I guess the limited number it is partly due to the restraints on keywords searched and inclusion criteria. For instance, the authors choose to go with Roger Ulrich as a key reference but exclude other research relating more to the built environment domains or include ‘biophilia’ or ‘healing spaces’ as keywords. Therefore, I think it would strengthen the paper to; a) address these specific choices of keywords more thoroughly, and b) address the consequences of these choices on keywords in a more critical reflection in the end of the paper.

Thank you.

Reviewer 2 Report

This is essentially a systematic review on the topic of nature-based pain interventions. The paper looks at acute and chronic pain and fits different models of NBT around each; as a result, the reader does gain knowledge of existing theoretical models as well as what support- however minimal- exists for each. In the end, the papers cited provide a framework but not sufficient support. I do not see any significant issues with this short review. I think it will provide readers an overview of this topic, which lies a bit outside mainstream biomedical care for pain.

Reviewer 3 Report

Dear Authors,

Thank you for a great effort spending on this paper. Please find my comments below:

1. Abstract: Line 21-22 This is too strong message. “Without this development, research on nature-based interventions and pain will continue to use proxy concepts for measurement and result in misrepresented findings”

Please rewrite in form of recommendations. For example:  With the Theoretical development, research on nature-based…. May provide the conceptual measurement………

2. Keywords: there are many terms in keywords that nor presented or related to the abstract such as nursing theory, biophilia hypothesis. It’s important that the abstract should offer messages linked to the keywords (directly or indirectly).Please reconsider the keywords or adding more information in the abstract

3. Materials and methods: need more details to explain the process for example ‘who doing what’ in each process. What were tools used to critique the literature. Please provide the flowchart diagram of integrative review (selection process based on PRISMA)

4. What are six steps and how the process was conducted, by who, any tools were used in these six steps.

5. Table 3- you should present the authors of each study in the table

6. The Theory summary section provide general information of the two theories (SRT and ART, it would be much better if you could link these theories back to the literature review outcomes. This section is quiet unflow and irrelevant to each other – literature review and then theory. Please revised this section.

7. The same as comment for the abstract Line 459-460 vs line 21-22—please rewrite the se messages.

8. Discussion may need to be more succinct; you can remove some parts that repeated the theory summary section.

Good luck

Round 2

Reviewer 3 Report

Dear Authorts,

Thnak you for the revised version with a clear response, appreciated.

I looking forward to this article.

Good luck,